# Design and Evaluation of ^223^Ra-Labeled and Anti-PSMA Targeted NaA Nanozeolites for Prostate Cancer Therapy—Part II. Toxicity, Pharmacokinetics and Biodistribution

**DOI:** 10.3390/ijms22115702

**Published:** 2021-05-27

**Authors:** Anna Lankoff, Malwina Czerwińska, Rafał Walczak, Urszula Karczmarczyk, Kamil Tomczyk, Kamil Brzóska, Giulio Fracasso, Piotr Garnuszek, Renata Mikołajczak, Marcin Kruszewski

**Affiliations:** 1Centre for Radiobiology and Biological Dosimetry, Institute of Nuclear Chemistry and Technology, Dorodna 16, 03-195 Warsaw, Poland; m.wasilewska@ichtj.waw.pl (M.C.); K.Brzoska@ichtj.waw.pl (K.B.); m.kruszewski@ichtj.waw.pl (M.K.); 2Department of Medical Biology, Institute of Biology, Jan Kochanowski University, Uniwersytecka 7, 24-406 Kielce, Poland; 3Centre of Radiochemistry and Nuclear Chemistry, Institute of Nuclear Chemistry and Technology, Dorodna 16, 03-195 Warsaw, Poland; r.walczak@ichtj.waw.pl; 4National Centre for Nuclear Research, Radioisotope Centre POLATOM, Sołtana 7, 05-400 Otwock, Poland; urszula.karczmarczyk@polatom.pl (U.K.); Kamil.Tomczyk@polatom.pl (K.T.); Piotr.Garnuszek@polatom.pl (P.G.); renata.mikolajczak@polatom.pl (R.M.); 5Department of Medicine, University of Verona, 37129 Verona, Italy; giulio.fracasso@univr.it; 6Department of Molecular Biology and Translational Research, Institute of Rural Health, Jaczewskiego 2, 20-090 Lublin, Poland

**Keywords:** PSMA-targeted radioligand therapy, prostate cancer, radium-223, D2B antibodies, zeolite nanoparticles, toxicity, pharmacokinetics, biodistribution

## Abstract

Metastatic castration-resistant prostate cancer (mCRPC) is a progressive and incurable disease with poor prognosis for patients. Despite introduction of novel therapies, the mortality rate remains high. An attractive alternative for extension of the life of mCRPC patients is PSMA-based targeted radioimmunotherapy. In this paper, we extended our in vitro study of ^223^Ra-labeled and PSMA-targeted NaA nanozeolites [^223^RaA-silane-PEG-D2B] by undertaking comprehensive preclinical in vitro and in vivo research. The toxicity of the new compound was evaluated in LNCaP C4-2, DU-145, RWPE-1 and HPrEC prostate cells and in BALB/c mice. The tissue distribution of ^133^Ba- and ^223^Ra-labeled conjugates was studied at different time points after injection in BALB/c and LNCaP C4-2 tumor-bearing BALB/c Nude mice. No obvious symptoms of antibody-free and antibody-functionalized nanocarriers cytotoxicity and immunotoxicity was found, while exposure to ^223^Ra-labeled conjugates resulted in bone marrow fibrosis, decreased the number of WBC and platelets and elevated serum concentrations of ALT and AST enzymes. Biodistribution studies revealed high accumulation of ^223^Ra-labeled conjugates in the liver, lungs, spleen and bone tissue. Nontargeted and PSMA-targeted radioconjugates exhibited a similar, marginal uptake in tumour lesions. In conclusion, despite the fact that NaA nanozeolites are safe carriers, the intravenous administration of NaA nanozeolite-based radioconjugates is dubious due to its high accumulation in the lungs, liver, spleen and bones.

## 1. Introduction

Metastatic castration-resistant prostate cancer (mCRPC) is a progressive and incurable disease with poor prognosis for patients [1]. Despite several options for the treatment of mCRPC, such as taxanes-based chemotherapy, second-generation antiandrogens, poly-(ADP-ribose)-polymerase (PARP) inhibitors, immunotherapy with sipuleucel-T and therapy with the bone-seeking radium-223 dichloride (Xofigo^®^), the median survival is ~30 months with a 30% 5-year survival rate.

A major source of mortality in mCRPC are bone metastases [2]. An attractive alternative concept for extension of the life of mCRPC patients with acceptable side effects is the PSMA-based targeted radionuclide therapy (TRT). The PSMA-based TRT uses a targeting vector (anti-PSMA antibody-based molecules or small molecule enzyme inhibitors) attached directly to radionuclide or indirectly by radionuclide carriers [3]. The lessons learned from 20 years of experience with PSMA vectors labeled with beta emitters have shown that this type of therapy was associated with ≥50% reduction in PSA level. The therapy allowed for efficient treatment of large tumors but has been suboptimal for the eradication of small cell clusters. Moreover, up to 30% of patients have never responded to this therapy, developed resistance, or suffered from serious hematological toxicity. The limitations of beta-emitter-based therapies may be linked to insufficient dose delivery to the tumor and the low linear-energy transfer of β-particles [4,5].

Recent studies have shown that PSMA-based targeted radionuclide therapy (TRT) with alpha emitters appears to be a more attractive therapeutic alternative than PSMA-based TRIT with beta emitters due to the release of a significantly higher amount of energy over a far shorter track [6]. Therapeutic efficiency and safety of targeted alpha emitters therapy of mCRCP have been tested in clinical studies, including approximately 250 patients treated with PSMA small molecule inhibitor PSMA-617, labeled with actinium-225, that has shown a significant anti-tumor efficacy [7]. Moreover, full biochemical responses were observed in patients who had not responded to therapy with [^177^Lu]Lu-PSMA-617. A disadvantage of this therapy has been the high number of patients who discontinued treatment because of severe xerostomia [8].

Despite promising results, the main impediment to the widespread use of alfa emitters in targeted therapy is the high cost of their production and low availability, which is currently sufficient to conduct only preclinical studies and a limited number of clinical trials [9,10]. The exception is ^223^Ra radionuclide, which can be produced in large amounts and relatively inexpensively [11]. ^223^Ra emits particles with a linear energy transfer (LET) of ~28 MeV over a range of ≤100 mcm [12]. Therefore, it has the potential to deliver therapeutically relevant doses from a small amount of injected activity. Another attractive property of ^223^Ra radionuclide is relatively long half-live (*t*_1/2_ = 11.43 days) that gives enough time for preparation, quality control, and shipment of the radiopharmaceutical. ^223^Ra in its simple form of radium dichloride (^223^RaCl_2,_ Xofigo^®^) is the only α-particle emitting therapeutic agent approved by the U.S. Food and Drug Administration (FDA) for systemic treatment of mCRPC patients with symptomatic osseous metastases [13]. Nevertheless, despite the established clinical efficacy and safety, as well as availability, ^223^Ra has not yet found application in receptor-targeted therapy because of the lack of effective bifunctional chelating agents [14]. Recently, Abou et al. [15] developed a biologically stable radiocomplex of [^223^Ra]Ra^2+^ with the 18-membered bis-picolinate diazacrown macrocyclic chelator MACROPA and demonstrated its efficient conjugation to a single amino acid β-alanine and to a glutamate–urea–glutamate (DUPA) PSMA-targeting vector. Unfortunately, upon conjugation of MACROPA to DUPA, stability of radiocomplex was lost. Alternatively, several attempts have been made to incorporate ^223^Ra into nanoparticle constructs such as liposomes [16,17], magnetite nanoparticles [18], CaCO_3_ microparticles [19], hydroxyapatites and titanium dioxide nanoparticles [20], barium sulfate nanoparticles [21] and nanozeolites [22,23]. Recently, our group successfully designed and evaluated in vitro the ^223^Ra-labeled and anti-PSMA targeted NaA nanozeolites (^223^RaA-silane-PEG-D2B) for prostate cancer therapy [24]. In this paper, we extended our previous in vitro study by undertaking comprehensive research concerning the toxicity and biodistribution of ^223^RaA-silane-PEG-D2B in normal (RWPE-1 and HPrEC) and cancer (LNCaP C4-2 and DU-145) prostate cells, BALB/c and LNCaP C4-2 xenograft-bearing BALB/c Nude mice.

## 2. Results

### 2.1. Synthesis and Physicochemical Characterization of ^223^RaA-Silane-PEG-D2B Radioconjugate

Synthesis and physicochemical characterization of ^223^RaA-silane-PEG-D2B and its derivatives were described in detail by Czerwińska et al. [24]. Briefly, these compounds were synthesized from aluminosilicate gel using the hydrothermal method. In the first step, the NaA zeolite nanocarrier was prepared. Next, the NaA nanocarrier was labeled with ^223^Ra^2+^ radionuclide by exchanging Na^+^ for ^223^Ra^2+^ cations, and the surface was modified with silane–PEG groups by siloxane bonds formation [^223^RaA-silane-PEG and NaA-silane-PEG]. In the last step, an anti-PSMA D2B antibody was conjugated to the nanocarrier via the 2-iminothiolane/m-maleimidobenzoyl- N-hydroxysuccinimide ester coupling method [^223^RaA-silane-PEG-D2B and NaA-silane-PEG-D2B] (Figure 1). The obtained compound was in the form of nanocrystals with a regular cubic-like shape, an average nominal diameter of ~120 nm, and an average hydrodynamic diameter of ~200 nm. The radiolabeled compounds [^223^RaA-silane-PEG-D2B and ^223^RaA-silane-PEG] had a specific activity of 0.65 MBq mg^−1^. It was estimated that ~200 silane-PEG groups and ~50 D2B molecules were coupled with one NaA zeolite molecule.

### 2.2. Analysis of Cell Death by the Annexin/Propidium Iodide Assay

The annexin/propidium iodide assay was used to investigate the effect of 50 mcg mL^−1^ NaA-silane-PEG and NaA-silane-PEG-D2B on the induction of cell death in LNCaP C4-2, DU-145, RWPE-1 and HPrEC cells following 94 h of incubation. Our results revealed that these nanocarriers induced a statistically significant increase of apoptosis and necrosis in HPrEC cells (Figure 2). NaA-silane-PEG elevated the percentage of early apoptotic and late apoptotic/necrotic HPrEC cells to 1.7-fold (1.9% vs. 1.1%) and to 1.5-fold (12.1% vs. 8.1%) of the control cells, respectively. NaA-silane-PEG-D2B increased the percentage of early apoptotic and late apoptotic/necrotic HPrEC cells to 2.3-fold (2.5% vs. 1.1%) and to 1.7-fold (13.5% vs. 8.1%) of the control cells, respectively. No significant difference in percentage of apoptotic and necrotic cells was observed in LNCaP C4-2, DU-145 and RWPE-1 cells.

### 2.3. Analysis of Apoptosis by Caspase 3/7 Assay

To confirm the results obtained with the annexin/propidium iodide assay, we determined the percentage of caspase 3/7 positive cells, representing apoptotic cells. Our results were nearly identical to those observed by the annexin/propidium iodide assay and revealed that NaA-silane-PEG and NaA-silane-PEG-D2B significantly induced apoptosis of HPrEC cells, while no differences were observed in LNCaP C4-2, DU-145 and RWPE-1 cells (Figure 3). NaA-silane-PEG significantly elevated the percentage of apoptotic HPrEC cells to 1.8-fold (2.5% vs. 1.4%) of the control cells, whereas NaA-silane-PEG-D2B significantly increased the percentage of apoptotic cells to 2.6-fold (3.7% vs. 1.4%) of the control cells.

### 2.4. Gene Expression Profiling

NaA-silane-PEG and NaA-silane-PEG-D2B were analyzed by real-time PCR for their effects on the expression of 84 key genes involved in autoimmune and inflammatory responses and 84 key genes involved in NF-κB signaling in LNCaP C4-2, DU-145, RWPE-1 and HPrEC cells. The detailed results are presented in Appendix A, while the number of genes significantly affected by these nanocarriers is presented in Venn diagrams (Figure 4).

For NaA-silane-PEG, only 2 out of 84 genes involved in autoimmune and inflammatory responses were deregulated in LNCaP C4-2 cells, 3 genes in DU-145 cells, 5 genes in RWPE-1 cells and 8 genes in HPrEC cells (Figure 4A). For NaA-silane-PEG, only 1 out of 84 genes involved in NF-κB signaling was affected in LNCaP C4-2 cells, 11 genes in DU-145 cells, 1 gene in RWPE-1 cells and 12 genes in HPrEC cells (Figure 4B). As shown in Figure 4C, NaA-silane-PEG-D2B deregulated 5 out of 84 genes involved in autoimmune and inflammatory responses in LNCaP C4-2 cells, 5 genes in DU-145 cells, 15 genes in RWPE-1 cells and 27 genes in HPrEC cells. For NaA-silane-PEG-D2B, 6 out of 84 genes involved in NF-κB signaling was affected in LNCaP C4-2 cells, 10 genes in DU-145 cells, 12 genes in RWPE-1 cells and 38 genes in HPrEC cells (Figure 4D). The overlap in expression of genes between cell lines was very small. Since the vast majority of genes reported in Figure 4 presented very low fold changes (<1.5) despite statistical significance, we focused on further analysis of the statistically significant results with cut-off value for gene expression fold changes >1.5. The results revealed that NaA-silane-PEG deregulated expression of 4 genes in DU-145 cells, 1 gene in RWPE-1 cells and 3 genes in HPrEC cells (Figure 5). There were no significant differences in gene expression in LNCaP C4-2 cells. However, NaA-silane-PEG-D2B deregulated expression of 2 genes in LNCaP C4-2 cells, 3 genes in DU-145 cells and 7 genes in RWPE-1 cells (Figure 6A). The effects of NaA-silane-PEG-D2B on gene expression were much more pronounced in primary HPrEC cells. As shown in Figure 6B, NaA-silane-PEG-D2B upregulated expression of 22 genes and downregulated expression of 7 genes.

### 2.5. Toxicity In Vivo

The nanocarrier and radioconjugate in vivo toxicity was tested on a homogeneous group of BALB/c mice, using unlabeled (NaA-silane-PEG and NaA-silane-PEG-D2B) and radiolabeled (^223^RaA-silane-PEG and ^223^RaA-silane-PEG-D2B) conjugates, 1 and 7 days after a single intravenous injection of the dose of c.a 6.4–8 mcg kg^−1^ b.w. and 49–51 kBq per mouse. The injected doses were well tolerated by all animals without treatment-related lethality. Moreover, no changes were observed in body weight, blood morphology, biochemical parameters and during histological examination of the liver, kidneys and femur 1 day after injection for all compounds, as compared to the control group (PBS). However, 7 days of exposure to ^223^RaA-silane-PEG and ^223^RaA-silane-PEG-D2B induced toxic effects, manifested by body weight loss (Figure 7a) and an abnormally low level of platelets, significantly different from the control mice and mice exposed to unlabeled NaA-silane-PEG and NaA-silane-PEG-D2B compounds (Figure 7c). In addition, the increased serum concentrations of ALT and AST enzymes were observed in mice exposed to all compounds, as compared to the control group (Figure 7b). Furthermore, the results showed that mice exposed to ^223^RaA-silane-PEG and ^223^RaA-silane-PEG-D2B had significantly lower red blood cell (RBC) and white blood cell (WBC) counts, as well as hematocrit (HCT) and hemoglobin (HGB) levels, at 7 days after injection, compared to the results at 1 day after injection (Figure 7d,e, Table 1 and Appendix B (Table A3, Table A4, Table A5, Table A6 and Table A7).

The mean value of RBC (red blood cell) counts never decreased below the lower reference limit (6.93 × 10^3^ mcL) [23], but 7 days after injection, the concentration of platelets (Figure 7c) decreased for both radiolabeled compounds, and this value was significantly different with respect to the 1 day after injection value (*p* = 0.030). At the same time, we observed the significant decrease of HCT (hematocrit), below the lower reference limit (42.1%), and HGB (hemoglobin). Particular attention should be paid to the strong leukopenia in both groups of animals (^223^RaA-silane-PEG and ^223^RaA-silane-PEG-D2B) 7 days after injection. The mean value of WBC significantly decreased below the lower reference limit (3.48 G L^−1^) and was 10-fold lower that the value at 1 day. Detailed analysis of the individual fractions of WBC revealed significant differences: neutrophil (NEUT) counts—over 20-fold decrease, lymphocyte (LYMPH) counts—10-fold decrease and monocyte (MONO) counts—17-fold decrease.

Examination of hematoxylin-eosin-stained mice tissues collected 1 day after injection revealed no changes in the liver, kidneys, and bone marrow for unlabeled (NaA-silane-PEG and NaA-silane-PEG-D2B) and radiolabeled (^223^RaA-silane-PEG and ^223^RaA-silane-PEG-D2B) compounds. Additionally, histopathological examination of these tissues collected 7 days after injection from mice treated with unlabeled NaA-silane-PEG and NaA-silane-PEG-D2B compounds showed no changes, as compared with tissues from the control mice. However, 7 days of exposure to ^223^RaA-silane-PEG and ^223^RaA-silane-PEG-D2B induced bone marrow fibrosis, without any changes in the liver and kidneys (Figure 8).

### 2.6. Biodistribution of ^223^RaA-Silane-PEG and ^223^RaA-Silane-PEG-D2B in Healthy BALB/c and Tumour Bearing BALB/c Nude Mice

Biodistribution studies of ^223^RaA-silane-PEG and ^223^RaA-silane-PEG-D2B in BALB/c mice were planned on the basis of the results dealing with pharmacokinetics and the biodistribution of ^133^BaA-silane-PEG and ^133^BaA-silane-PEG-D2B in BALB/c mice (Appendix B
Table A1, Figure A1). ^133^Ba radionuclide is γ-emitter and is particularly feasible as a diagnostic match to the therapeutic α-emitter ^223^Ra radionuclide [21].

In the first set of experiments, the biodistribution of ^223^RaA-silane-PEG and ^223^RaA-silane-PEG-D2B was evaluated in BALB/c mice 1 and 7 days after injection. Mice were exposed to a dose of c.a 8 mcg kg^−1^ b.w. and 38 and 48 kBq per mouse, respectively. Side-by-side comparison is shown in Table 2.

To investigate the efficacy of ^223^RaA-silane-PEG-D2B penetration into the prostate tumor, we used BALB/c Nude mice with a subcutaneous prostate tumor xenograft derived from the PSMA-positive LNCaP C4-2 prostate cancer cell line. Biodistribution studies were conducted 4 h, 24 h, 72 h and 7 days after injection. ^223^RaA-silane-PEG was used as a control. Both radioconjugates were applied at in average dose of c.a 12 mcg kg^−1^ b.w. and 50 kBq per mouse. Side-by-side comparison is shown in Table 3.

Statistically significant differences were observed between radioactivity (%ID g^−1^) in the lungs, spleen, bones and blood of healthy BALB/c mice and tumour bearing BALB/c Nude mice. Radioactivity of ^223^RaA-silane-PEG and ^223^RaA-silane-PEG-D2B in the lungs and bones of BALB/c mice was 2-fold lower than in tumour-bearing mice. Both radioconjugates were excreted more slowly by tumour-bearing mice than healthy mice. However, ^223^RaA-silane-PEG-D2B accumulated at significantly higher levels in the spleen, liver and lungs of tumour-bearing mice at 7 day, as compared to the ^223^RaA-silane-PEG-injected animals. The bones exhibited similar radioactivity in both ^223^RaA-silane-PEG-D2B- and ^223^RaA-silane-PEG-exposed animals. Surprisingly, we observed very low uptake of ^223^RaA-silane-PEG-D2B in tumour lesions (below 1 %ID g^−1^), despite the presence of a targeting antibody. The value of accumulation was higher for ^223^RaA-silane-PEG than for ^223^RaA-silane-PEG-D2B, and this was related to the blood flow through the tumour blood vessels but not to specific localization. The lower tumour accumulation value for ^223^RaA-silane-PEG-D2B corresponded to lower blood concentration of the radioactivity. After 24 h, both ^223^RaA-silane-PEG and ^223^RaA-silane-PEG-D2B were still present in the liver and spleen with high %ID g^−1^.

## 3. Discussion

In the present study, we aimed to extent our previous findings, which revealed that the radiobioconjugate ^223^RaA-silane-PEG-D2B was very stable in human blood serum in vitro, bound specifically and internalized into PSMA-expressing LNCaP C4-2 cells, as well as demonstrating potent radiotoxicity in LNCaP C4-2 cells [24].

Since one of the major challenges in the application of nanomaterials as carriers to deliver therapeutics for targeted therapy is their safety, the first goal of this study was to evaluate the in vitro and in vivo toxicity of an antibody-free nanocarrier (NaA-silane-PEG) and antibody-functionalized nanocarrier (NaA-silane-PEG-D2B). Our results concluded that both nanocarriers express no cytotoxic activity against established normal and cancer cell lines and very low cytotoxic activity in extremely sensitive primary cells, which are ill adapted to the two-dimensional culture and changes in their environment [26]. These results are concordant with our previous findings, which showed that NaA-silane-PEG-D2B had no effect on the metabolic activity of LNCaP C4-2 and DU145 cells at a dose of 100 mcg mL^−1^ following 96 h of treatment time [24]. We also previously reported that NaA nanozeolites coated with long PEG molecules (MW1000 and MW2000), as well as unmodified NaA nanozeolite, were nontoxic in HeLa and HEK239 cells [27]. Moreover, no significant cytotoxic activity was also reported for nanozeolites A and Y in macrophages, epithelial and endothelial cells [28], and for pure silica nanozeolites in HeLa cells [29].

Apart from the determination of cytotoxicity of nanocarriers, the assessment of their immunotoxicity is an important component of safety evaluation [30]. It is well known that nanoparticles can interact with various components of the immune system, potentially leading to undesirable immunosuppression, hypersensitivity, immunogenicity, and autoimmunity, involving both innate and adaptive immune responses [31]. Moreover, previous studies demonstrated that accumulation of nanoparticles in the body can modulate pro-inflammatory cytokine production, which in turn is involved in the regulation of cellular events in prostate carcinogenesis and metastasis [32]. One of the valuable tools in screening nanoparticle immunotoxicity is the measurement of expression of immune-related genes [33]. Therefore, we conducted a comprehensive analysis of immune gene profiling, which involved an analysis of the expression of 84 genes involved in the NF-κB signaling, which plays a causative role in inflammatory processes, controls the transcription of cytokines and genes that regulate various aspects of innate and adaptive immune responses [34] and the expression of 84 genes involved in the inflammatory response and autoimmunity. We found that antibody-free nanocarrier and antibody-functionalized nanocarrier triggered very week response of a small number of genes in vitro. Most of the genes deregulated by NaA-silane-PEG and NaA-silane-PEG-D2B encode proteins involved in both stimulation and suppression of pro-inflammatory responses [35,36,37,38,39]. Simultaneous deregulation of genes encoding proteins that are key to both pro-inflammatory and anti-inflammatory responses in the same cell line does not allow a definitive statement about the immunotoxicity of these carriers. Nevertheless, we cannot exclude the possibility that prolonged exposure to these carriers due to their accumulation in the body may contribute to the development of inflammation, responsible for spread of prostate cancer [32,38].

Since the in vitro behavior of nanocarriers frequently does not correlate with their in vivo responses [40], we performed in vivo toxicity studies in a homogeneous group of BALB/c mice. Our results revealed no changes in body weight or the level of platelet and blood morphology 7 days after a single administration of NaA-silane-PEG and NaA-silane-PEG-D2B. However, we observed increased serum levels of aspartate aminotransferase (AST) and alanine aminotransferase (ALT), which are the most sensitive liver enzymes (Figure 7) [41]. We suppose that in our study the elevated concentration of these enzymes was a consequence of nanocarrier accumulation in the liver. These results are consistent with other studies that demonstrated elevated ALT and AST levels due to a high accumulation of different types of nanocarriers in the liver [42].

Taking into account no obvious symptoms of toxicity of both carriers revealed by in vitro and in vivo studies, we investigated biodistribution and radiotoxicity of ^223^Ra-labeled conjugates at 24 h and 7 days post injection in healthy BALB/c mice. Critical organs for ^223^RaA-silane-PEG and ^223^RaA-silane-PEG-D2B toxicity were the liver, spleen and the lungs. We assume that the observed accumulation of radioactivity in these organs was related to the phagocytosis of ^223^RaA-silane-PEG and ^223^RaA-silane-PEG-D2B by Kupffer cells in the liver, pulpa macrophages in the spleen and alveolar macrophages in the lungs. The highest accumulation of ^223^RaA-silane-PEG-D2B was observed in the spleen, which was 2-fold higher compared with ^223^RaA-silane-PEG. This might be due to the reduction of spleen mass, observed in our study. The result is in line with findings, which showed a radiation-induced progressive spleen volume decrease [43]. During the next six days, the accumulation in the spleen decreased to reach a similar value for both radiolabelled compounds. The increased accumulation of nanoparticle-based conjugates in organs of the reticuloendothelial system was reported by others in healthy mice [44,45]. In our opinion, the biodistribution pattern of ^223^RaA-silane-PEG and ^223^RaA-silane-PEG-D2B is regulated mainly by their hydrodynamic size, the presence of antibodies, and probably on the hydrophilicity of the particle surface. The ^223^RaA-silane-PEG-D2B molecules presented ~50 D2B antibodies on the coat, with a final size of nanoparticles ~250 nm, whereas the antibody-free ^223^RaA-silane-PEG molecules had a diameter ~196 nm [24]. It is well known that once nanomaterials are introduced into biological environments, such as blood, serum, or intracellular fluid, they adsorb on their surfaces the so-called “biocorona”, containing many proteins and lipids [46]. Bio-corona formation depends on physicochemical and biological properties of nanoparticles, which in turn influences their uptake by cells, biodistribution and circulation half-time [46,47].

In addition, we observed the high radioactivity level in bone tissue for both radioconjugates at 72 h post injection. We suppose that this effect was related to the leakage of ^223^Ra radionuclide and its decay products from nanozeolites, and to the natural affinity of ^223^Ra radionuclide with bones [48]. Our in vivo results are in contrast to the in vitro findings, which revealed only ~5% leakage of ^223^Ra and ~6% leakage of its decay products from bioconjugate in human blood serum after 12 days [24]. However, it should be noted that the in vitro stability results could not be transferred to in vivo models, where blood flow may easily dislocate the decay products from the surface of the nanozeolite particles and reduce the resorption probability [45,49]. Referring to the limited stability of radiolabeled nanozeolites, in our opinion, the suitability of NaA nanozeolite as the ^223^Ra delivery system is dubious from a clinical point of view. A comparison of the biodistribution of ^223^Ra-silane-PEG-D2B and ^223^Ra-silane-PEG with the biodistribution of ^223^RaCl_2_, based on data from the Alpharadin (Xofigo) Assessment Report [50], showed that accumulation of ^223^RaCl_2_ in the bones, liver and spleen was significantly lower than ^223^Ra-silane-PEG-D2B and ^223^Ra-silane-PEG (Appendix B
Figure A2, Table A2). The difference in the accumulation of the tested compounds in the small intestine, large intestine and kidney were not statistically significant. This is most likely related to the size of the tested molecules compared with ^223^RaCl_2_. The much larger diameter of ^223^Ra-silane-PEG-D2B and ^223^Ra-silane-PEG suggest that retention of these molecules in the liver or spleen may be higher than ^223^Ra ions alone. On the other hand, we cannot rule out that the tested compounds are not stable in the living organism, and thus dissociation of ^223^Ra from the nanoconjugates and redistribution may occur. This is evidenced by the gradually increasing accumulation of radioactivity in the bone.

The increased accumulation of ^223^Ra-labeled conjugates in bone tissue led to bone marrow fibrosis, observed 7 days post injection, which was accompanied by an abnormally low level of platelets and white blood cells. Similar findings with other radio-conjugates were also observed in preclinical in vivo studies and in patients receiving targeted radioligand therapy [51,52]. In addition, we observed the increased serum concentrations of ALT and AST enzymes. This observation is probably a reflection of the increased accumulation of ^223^RaA-silane-PEG-D2B and ^223^RaA-silane-PEG in the liver. However, we did not detect any sign of histological damage in this tissue. These observations were accompanied by a significant decrease in the body weight of the animals in both groups of mice (^223^Ra-silane-PEG and ^223^Ra-silane-PEG-D2B). Therefore, due to animal welfare considerations, no further follow-up beyond 7 days was conducted. The significant toxicity observed after administration of radioactivity in the formulations of approximately 2 MBq/kg, was similar to the acute toxicity results reported in the literature in preclinical studies of ^223^RaCl_2_ in mice, where the NOAEL for ^223^RaCl_2_ was found to be <1250 kBq/kg bw [50,53].

To investigate the efficacy of ^223^RaA-silane-PEG-D2B in its effective delivery to the prostate tumor, we used BALB/c Nude mice with a subcutaneous prostate tumor xenograft derived from the PSMA-positive LNCaP C4-2 prostate cancer cell line. Surprisingly, we observed very low uptake of ^223^RaA-silane-PEG-D2B to tumour mass (below 1 %ID g^−1^), despite the presence of a targeting antibody. This observation clearly showed that the injected ^223^RaA-silane-PEG-D2B was not preferentially transported to the prostate tumor xenograft. The reasons for the lack of effective delivery to the prostate tumor are not fully understood, especially since our in vitro studies demonstrated the highly specific binding of ^223^RaA-silane-PEG-D2B at the PSMA-expressing LNCaP C4-2 cell surface and its fast cellular internalization. Moreover, the D2B antibody alone has excellent in vivo tumor targeting characteristics, showing the high LNCaP xenograft uptake at doses from 0.1 to 3 mcg/mouse (~50 %ID g^−1^) [54]. The dose of D2B antibody used in our study was within this range (~1 mcg/mouse), indicating that this factor was probably not the reason for low tumor uptake. The most likely explanation for the observed low tumor deposition of ^223^RaA-silane-PEG-D2B is the relatively high hydrodynamic diameter of ^223^RaA-silane-PEG-D2B (~250 nm) and low Enhanced Permeability and Retention effect (EPR), which is responsible for extravasation and retention of nanocarriers in tumors [55]. Though tumors growing in the subcutaneous microenvironment have a functional pore cut-off size ranging from approximately 10 nm to 1000 µm [55,56,57], the size of endothelial fenestrae and vascular permeability in tumors is highly heterogeneous and depends on tumor type, development and growth [58]. It was reported that slow-growing tumors, such as the prostate tumor, have a decreased EPR effect and are usually difficult to treat with nanomedicine [59]. Similar findings were reported for different tumor xenografts [45,60,61].

## 4. Materials and Methods

### 4.1. Reagents

All chemicals were of analytical grade and used without further purification. Aluminum isopropoxide (≥98% purity) as Al source, LUDOX CL-X colloidal silica (45% suspension in water) as Si source, sodium hydroxide (≥97% purity) as cation source, and tetramethylammonium hydroxide (TMAOH) solution (25 wt.% in water) as template were purchased from Sigma-Aldrich. Deionized water was obtained from Merck Millipore equipment (Saint Louis, MO, USA). All above reagents were used for synthesis of NaA zeolite powders. Etoxy silane functionalized polyethylene glycol (silane-PEG, 5000 Da) from Nanocs (Boston, MA, USA) and ethanol (96% and 99.8% in water) from POCH (Gliwice, Poland) were used for the silanization process. 2-iminothiolane (2-IT), sodium bicarbonate (NaHCO_3_), ethylenediaminetetraacetic acid (EDTA), glycine, Ellman’s reagent, m-maleimidobenzoyl-N-hydroxysuccinimide ester (MBS) and dimethylformamide (DMF) from Sigma-Aldrich (Lenexa, KS, USA) were used for the conjugation of anti-PSMA monoclonal antibodies (D2B) to silane-PEG-nanozeolite. Radionuclide of ^223^Ra was isolated by a radiochemical separation from the ^227^Ac source obtained from the Institute for Transuranium Elements (Karlsruhe, Germany) in the amount of 3 MBq. Radionuclide of ^133^Ba was supplied from the Radioisotope Centre POLATOM (Otwock, Poland).

### 4.2. Synthesis and Physicochemical Properties of ^223^RaA-Silane-PEG-D2B Radioimmunoconjugates

The synthesis and physicochemical properties of ^223^RaA-silane-PEG-D2B radioconjugate were described in detail by Czerwińska et al. [24]. Briefly, the NaA nanozeolites were synthesized using the hydrothermal method. Sodium hydroxide was dissolved in distilled water, and afterwards, TMAOH was added. This sodium hydroxide solution was divided into two equal volumes, and aluminum isopropoxide was added to one half of the solution. Silicate solution was prepared by dissolving colloidal silica in the other half of the solution and was combined with the aluminate solution to obtain a gel. Stirring was performed in an ice bath for 96 h (aging time). After this time, the mixture was placed in a high-pressure autoclave and heated at 100 °C for 24 h. Finally, the obtained material was washed and dried. In order to remove the template, the resulting product was calcined for 72 h at 600 °C. The ^223^Ra-labeled NaA nanozeolite was prepared by exchanging Na^+^ for ^223^Ra^2+^ cations in RaCl_2_ solution (activity ~0.6 MBq). The ^133^Ba-labeled NaA nanozeolite was prepared by exchanging Na^+^ for ^133^Ba^2+^ cations. The functionalization of the surface was carried out by using silane coupling agent with three ethoxy groups and PEG molecules. The anti-PSMA monoclonal antibody D2B (IgG1) was prepared as previously described by Frigeiro et al. [62]. Conjugation of the D2B antibody with the obtained NaA-silane-PEG nanozeolite consisted of three steps. In the first step the antibody D2B-SH derivative was formed, using 2-iminothiolane (2-IT). In the next step, m-maleimidobenzoyl-N-hydroxysuccinimide ester (MBS) was added to the silane-PEG-NH_2_ functionalized nanozeolite. Finally, the obtained NaA nanozeolite-silane-PEG-MBS was conjugated with Abs-D2B-2-IT. In addition to the final compound ^223^RaA-silane-PEG-D2B, its three derivatives, NaA nanozeolite modified with silane-PEG groups [NaA-silane-PEG], NaA nanozeolite modified with silane-PEG groups and functionalized with anti-PSMA D2B antibodies [NaA-silane-PEG-D2B] and NaA nanozeolite labeled with ^223^Ra radionuclide and modified with silane-PEG groups [^223^RaA-silane-PEG], were synthesized and characterized.

### 4.3. Cell Cultures

The human epithelial prostate carcinoma cell line DU-145 (ATCC^®^HTB-81™), the human lymph node prostate carcinoma cell line LNCaP C4-2 (ATCC^®^ CRL-3314^™^), the human epithelial prostate normal cell line (ATCC^®^ CRL-11609™) and the human primary prostate epithelial cells HPrEC (ATCC^®^ PCS-440-010™) were purchased from the American Type Tissue Culture Collection (ATCC, Rockville, MD, USA) and maintained according to ATCC protocols. Briefly, DU-145 were cultured in EMEM medium supplemented with 10% FBS, 2 mM L-glutamine and 100 U mL^−1^ penicillin-streptomycin; LNCaP C4-2 were grown in RPMI 1640 medium containing 10% FBS, 2 mM L-glutamine, 10 mM Hepes, 40 mg ^L−1^ folic acid and 100 U mL^−1^ penicillin-streptomycin; RWPE-1 were cultured in Keratinocyte Serum Free Medium supplemented with 0.05 mg mL^−1^ bovine pituitary extract and 5 ng mL^−1^ human recombinant epidermal growth factor 0.05 mg mL^−1^; and HPrEC cells were grown in Prostate Epithelial Cell Basal Medium supplemented with 6 mM L-Glutamine, 1 mM epinephrine, 0.4% extract *p*, 0.5 ng/mL rh TGF-a, 100 ng mL^−1^ hydrocortisone hemisuccinate, 5 mg mL^−1^ rh^−1^ Insulin, 5 mg mL^−1^ apo-transferrin and 100 U/mL penicillin-streptomycin. The cell lines were maintained in an incubator at 37 °C with 5% CO_2_.

### 4.4. Analysis of Cell Death by the Annexin/IP Assay

Annexin/IP assay was performed according to the manufacturer’s instructions (BD Biosciences). Briefly, cells were plated at 80,000 cells per well in a 12-well plate, incubated overnight, and then treated with 50 mcg mL^−1^ of NaA-silane-PEG or NaA-silane-PEG-D2B for 96 h. After that time, cells were gently harvested by tripsin-EDTA, washed twice with cold PBS, and resuspended in 1× binding buffer at 1 × 10^6^ cells mL^−1^. Then, FITC-conjugated annexin V and PI were added to each sample. Samples were incubated for 15 min in the dark, and 400 mcL of 1× binding buffer was added to each tube. Samples were run on a LSR II flow cytometer (BD Biosciences, San Diego, CA, USA) using FACSDiva software.

### 4.5. Detection of Caspase 3/7 Activities

CellEvent™ Caspase-3/7 Green Flow Cytometry assay was performed according to the manufacturer’s instructions (Thermo Fisher, Vienna, Austria). Briefly, LNCaP C4-2, DU-145, RWPE-1 and HPrEC cells were plated at 50,000 cells per well in a 12-well plate, incubated overnight, and then treated with 50 mcg/mL of NaA-silane-PEG or NaA-silane-PEG-D2B for 96 h. After that time, cells were gently harvested by tripsin-EDTA and washed twice with cold PBS. Then, cells were incubated with CellEvent Caspase-3/7 Green Detection Reagent to a final concentration of 2 μM for 30 min. During the last 5 min of incubation, 1 µL of SYTOX AADvanced dead cell stain solution was added to each sample. Samples were run on a LSR II flow cytometer (BD Biosciences, San Diego, CA, USA) using FACSDiva software.

### 4.6. RNA Isolation, Reverse Transcription, and Real-Time PCR

LNCaP C4-2, DU-145, RWPE-1 and HPrEC cells were plated at 70,000 cells per well in a 12-well plate and allowed to adhere. After 24 h, cell culture media were removed and new culture media with 50 mcg mL^−1^ of NaA-silane-PEG or NaA-silane-PEG-D2B were added. After a 24 h treatment period, cells were harvested and immediately frozen in liquid nitrogen until RNA isolation. Total RNA was extracted from cell pellets using the ReliaPrep RNA Cell Miniprep System (Promega, Madison, WI, USA) according to manufacturer’s protocol. RNA concentration was measured using Quantus Fluorometer (Promega, Madison, WI, USA) and the QuantiFluor RNA System (Promega, Madison, WI, USA). RNA integrity was tested by agarose gel electrophoresis. For PCR array analysis, 1 mcg of total RNA was converted to complementary DNA (cDNA) in a 20-mcL reaction volume using RT^2^ First Strand Kit (Qiagen, Hilden, Germany). The cDNA was diluted with 91 mcL distilled water and used for expression profiling using the human NF-κB Signaling Targets PCR Array (Qiagen, Hilden, Germany, cat. no. PAHS-225Z) and the human Inflammatory Response and Autoimmunity PCR Array (Qiagen, Hilden, Germany, cat. no. PAHS-077Z) according to manufacturer’s instructions. Briefly, a total volume of 25 mcL of PCR reaction mixture, which included 12.5 mcL of RT^2^ SYBR Green/ROX qPCR Master Mix (containing HotStart DNA Taq polymerase, SYBR Green dye and the ROX reference dye), 11.5 mcL of double-distilled H_2_O, and 1 mcL of diluted template cDNA, was used for each primer set in each well of the PCR array. One technical replicate was performed for each sample. PCR amplification was carried out using a 7500 Real-Time PCR System (Thermo Fisher Scientific, Waltham, MA, USA) with an initial 10-min step at 95 °C followed by 40 cycles of 95 °C for 15s and 60 °C for 1 min. Relative gene expression was calculated using the ΔΔCt method with *ACTB*, *B2M*, *GAPDH*, *HPRT1*, and *RPLP0* as reference controls. Calculations were carried out using the Relative Quantification Software version 2019.2.7-Q2-19-build3 (Thermo Fisher Cloud, Thermo Fisher Scientific, Waltham, MA, USA). Statistical differences were examined by Student’s t-test with *p* < 0.05 considered to be statistically significant.

### 4.7. Experimental Animals

BALB/c male mice (5–6-week-old, mean body mass of 20 g) were purchased from the M. Mossakowski Institute of Experimental and Clinical Medicine at the Polish Academy of Sciences in Warsaw (Poland). BALB/c Nude male mice (7-week-old, mean body mass of 20 g) were purchased from the Charles River Laboratories (Sulzfeld, Germany). On arrival, animals were housed for five days in groups of five in standard cages (BALB/c) and IVS cages (BALB/c Nude) in the animal facility of the Radioisotope Centre POLATOM (Otwock, Poland). They were housed in a quiet room under constant conditions (22 °C, 50% relative humidity, 12-h light/dark cycles with dark period from 7 p.m. to 7 a.m.) with free access to standard food and water. Veterinarian staff and investigators observed the mice daily to ensure animal welfare and determine if humane endpoints were reached (e.g., hunched and ruffled appearance, apathy, ulceration, severe weight loss, tumor burden). Experimental procedures were carried out in conformity with the National Legislation and the Council Directive of the European Communities on the Protection of Animals Used for Experimental and Other Scientific Purposes (2010/63/UE) and the “ARRIVE guidelines for reporting animal research” [63]. The POLATOM protocol was approved by the 1st Local Animal Ethics Committee in Warsaw (authorization 429/2017, approval date 22 November 2017).

### 4.8. Experimental Groups and Treatment

Mice were divided into nine groups (G1–G9). G1 group: *n* = 35 BALB/c mice, treated with a single intravenous injection of ^133^BaA-silane-PEG (0.1 mL, 280–320 mcg, 185–209 kBq) into the lateral tail vein. Average dose was 12.3 ± 2.0 mcg kg^−1^ body weight. G2 group: *n* = 35 BALB/c mice, treated with a single intravenous injection of ^133^BaA-silane-PEG-D2B (0.1 mL, 280–320 mcg, 185–209 kBq) into the lateral tail vein. Average dose was 12.6 ± 1.5 mcg kg^−1^ body weight. G3 group: *n* = 10 BALB/c mice, treated with PBS (control group for G4-G7 groups taking part in toxicological studies). A 0.1 mL of PBS was injected into the lateral tail vein. G4 and G5 groups: *n* = 10 BALB/c mice (each) treated with NaA-silane-PEG and NaA-silane-PEG-D2B (0.1 mL, 150 mcg, 6.4 ± 0.68 mcg kg^−1^ body weight) in the lateral tail vein. G6 and G7 groups: *n* = 10 BALB/c mice (each) treated with ^223^RaA-silane-PEG (0.1 mL, 150 mcg, 49 kBq) and ^223^RaA-silane-PEG-D2B (0.1 mL, 150 mcg, 51 kBq) in the lateral tail vein. Average dose was 8.0 ± 1.5 mcg kg^−1^ body weight. G8 and G9 groups: *n* = 10 LNCaP C4–2 tumor-bearing NU/NU BALB/c mice (each) treated with ^223^RaA-silane-PEG (0.1 mL, 150 mcg, 38 kBq) and ^223^RaA-silane-PEG-D2B (0.1 mL, 150 mcg, 48 kBq) in the lateral tail vein. Average dose was 12.1 ± 2.8 mcg kg^−1^ body weight.

### 4.9. Pharmacokinetic Studies of ^133^BaA-Silane-PEG and ^133^BaA-Silane-PEG-D2B in BALB/c Mice

Mice were randomized into two groups, G1 and G2 (35 mice per group). Before injection, the ^133^Ba-labelled nanoparticles were diluted in PBS and then sonicated for 5 min to break up aggregates of micron-sized colloidal particles. At established time points (0.5 h, 1 h, 4 h, 24 h, 48 h and 72 h after injection), mice were anesthetized using isoflurane, weighed, and then put to death by cervical dislocation. Organs and tissues of interest were dissected, weighed and assayed for their radioactivity (Perkin Elmer Wallac 1470 Wizard Automatic Gamma Counter). The physiological distribution was then calculated and expressed in terms of the percentage of administrated radioactivity found in each of the selected organs or tissues per gram (%IDg^−1^). Pharmacokinetic analysis of the radiolabeled nanoparticles was based on the curves of the %IDg^−1^ accumulated in blood. Pharmacokinetic parameters were determined according to a one-phase exponential decay model.

### 4.10. Toxicity Studies of ^223^RaA-Silane-PEG and ^223^RaA-Silane-PEG-D2B in BALB/c Mice

Mice were randomized into five groups, from G3 to G7 (10 mice per group). Before injection, ^223^RaA-silane-PEG and ^223^RaA-silane-PEG-D2B were diluted in PBS and then sonicated for 5 min to break up aggregates of micron-sized colloidal particles. At 24 h and 7 days post injection, venous blood was drawn from the submandibular vein while the mouse was restrained but not anesthetized. Mice were put to death by cervical dislocation at 24 h and seven days post injection.

### 4.11. Measurements of Morphology Parameters

Venous blood (0.1 mL) was collected in K_2_EDTA-coated vials (Microvette^®^ 500 Sarstedt, Marfour, Polska, cat. no. 20.1341), immediately mixed using a rotary mixer, placed at +4 °C, and processed up to three hours after blood taking. The blood test was perform using a fluorescence flow cytometer XN-1000 V system (Sysmex XN-V Series, Milton Keynes, UK) with software dedicated for mouse species. Blood was analyzed for standard parameters: HTC (hematocrit), HGB (hemoglobin), WBC (white blood cells), EOS (eosinophils), BASO (basophils), NEUT (neutrophils), LYMPH (lymphocytes), MONO (monocytes), RBC (red blood cells), MCV (mean corpuscular volume), MCH (mean corpuscular hemoglobin), and MCHC (mean corpuscular hemoglobin concentration).

### 4.12. Measurements of Biochemical Parameters

The vein blood (ca 0.3 mL) was collected in blood collection serum tubes coated with a clot activator on the inner wall of the tubes (Microvette^®^ 200 Sarstedt, Marfour, Polska, cat. no. 20.1292). The blood test was carried out up to three hours after blood taking. For the biochemical profile, we used the Olympus AU680 Clinical Chemistry Analyzer and calorimetric method at 37 °C. We checked the following parameters and methods: enzymes (AST (aspartate aminotransferase), ALT (alanine aminotransferase)—IFCC Reference method with Pyrioxal Phosphate; Ap—IFCC Reference Method 2011), albumin—Bromocresol Green, urea—Urease/GLDH, creatinine—Jaffe IDMS and c-reactive protein.

### 4.13. Histopathology

The bone, liver and kidneys were fixed in 4% formalin in PBS. Organ samples were trimmed and embedded in paraffin. Then, histological sections were prepared, stained with hematoxylin-eosin and assessed under a light microscope.

### 4.14. Biodistribution of ^223^RaA-Silane-PEG and ^223^RaA-Silane-PEG-D2B in LNCaP C4-2 Tumor-Bearing BALB/c Nude Mice

LNCaP C4-2 cells were grown to 80–90% confluence before trypsinization and formulation in Matrigel™ Basement Membrane Matrix (Corning, Bedford, MA, US). BALB/c Nude mice were subcutaneously injected in the shoulder with 200 mcL of bolus containing a suspension of 5 × 10^6^ freshly harvested LNCaP C4-2 cells (100 mcL) in Matrigel™ (100 mcL) under anesthesia with 2% isoflurane. Mice were kept under pathogen-free conditions. Experiments were performed ~4-5 weeks later, when the tumor reached a volume of approximately 100 ± 40 mm^3^. Then, mice were randomized into two groups (10 mice per group): G8—treated with a single intravenous injection of ^223^RaA-silane-PEG and G9—treated with a single intravenous injection of ^223^RaA-silane-PEG-D2B. Mice were euthanized 4 h, 24 h, 72 h and 7 days after injection by an appropriate method and dissected. Selected organs and tissues were weighed and assayed for their radioactivity. Ex-vivo biodistribution was then calculated and expressed as the percentage of administrated radioactivity found in each organ or tissue per gram (%IDg^−1^).

### 4.15. Statistical Analysis

Statistical analysis of the in vitro data (with the exception of the PCR Array data) was performed using the Statistica 7.1 software (Stat Soft. Inc., Tulsa, OK, USA). The data were expressed as mean ± standard deviation (SD) of at least three independent experiments. Data were evaluated by the Kruskal-Wallis one-way analysis of variance on ranks (ANOVA) followed by the post-hoc Fisher’s test. Differences were considered statistically significant when the *p* value was less than < 0.05. Venn diagrams were drawn using the online web tool [64].

The results of in vivo biochemical parameters, concentration of platelets, and physiological distribution were expressed as a percentage of the dose administered per gram of tissue (%ID g^−1^) and presented in the form of an average with standard deviation (mean ± SD), with *n* representing the number of samples or animals per group. Data were statistically analyzed using GraphPad Prism version 8.0.0 for Windows and tested for normal distribution with the Kolmogorov–Smirnov test. In case of normal distribution, results were assessed by two-tailed, unpaired Student’s t-tests. Otherwise, results were assessed by two-way ANOVA. A *p* value of < 0.05 with two-tailed testing was considered statistically significant. For blood activity data, a one-phase exponential decay model was used to model the percentage of remaining activity (%ID/g) as a function of time post-injection (t):%ID g^−1^ = (%ID g^−1^ (0) − plateau) × exp(−K × T) + Plateau(1)
where:

%ID g^−1^ (0) is the %ID g^−1^ value when T (time) is zero

plateau is the %ID g^−1^ value at infinite times

K is the rate constants

Nonlinear least-squares regression was used to estimate the half-live of the exponential functions:T_1/2_ = ln(2)/K(2)

## Figures and Tables

**Figure 1 ijms-22-05702-f001:**
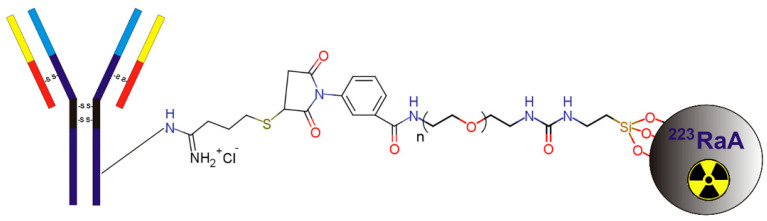
Scheme of ^223^Ra-labeled nanozeolite particle functionalization with silane-PEG group and anti-PSMA D2B antibody.

**Figure 2 ijms-22-05702-f002:**
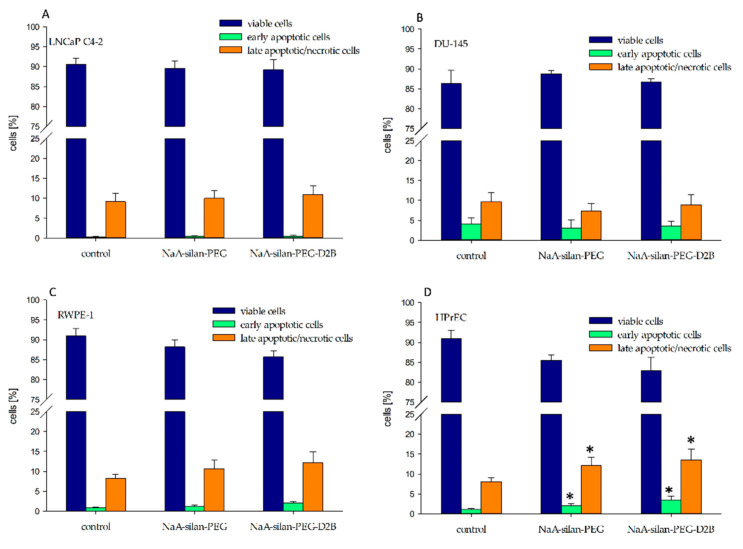
The effect of NaA-silane-PEG and NaA-silane-PEG-D2B on the proportion of viable cells (annexin V-negative and propidium iodide-negative cells), early apoptotic cells (annexin V-positive cells and propidium iodide-negative cells) and late apoptotic/necrotic cells (annexin V-positive and propidium iodide-positive cells). LNCaP C4-2 cells (**A**), DU-145 cells (**B**), RWPE-1 cells (**C**) and HPrEC cells (**D**) were exposed to 50 mcg mL^−1^ of NaA-silane-PEG or NaA-silane-PEG-D2B for 96 h. Data are expressed as the mean ± standard deviation (SD) from three independent experiments. * *p* < 0.05 versus control group.

**Figure 3 ijms-22-05702-f003:**
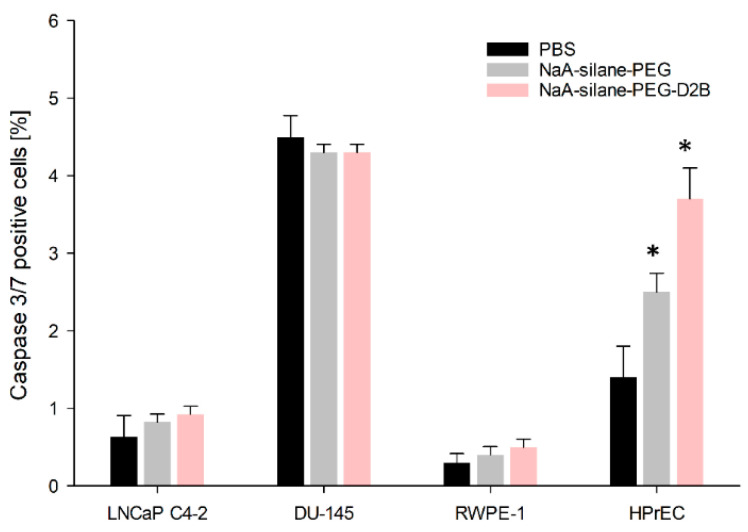
The effect of NaA-silane-PEG and NaA-silane-PEG-D2B on the proportion of caspase 3/7 positive cells. LNCaP C4-2 cells, DU-145 cells, RWPE-1 cells and HPrEC cells were exposed to 50 mcg mL^−1^ of NaA-silane-PEG or NaA-silane-PEG-D2B for 96 h. Data are expressed as the mean ± standard deviation (SD) from three independent experiments. * *p* < 0.05 versus control group.

**Figure 4 ijms-22-05702-f004:**
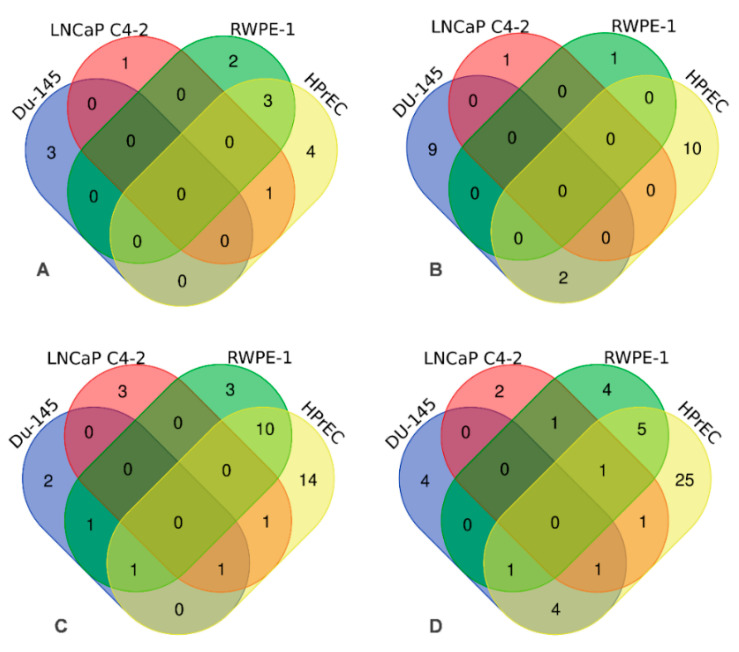
Venn diagrams depicting number of genes significantly affected (*p* < 0.05) by NaA-silanePEG and NaA-silane-PEG-D2B nanocarriers in LNCaP C4-2, DU-145, RWPE-1 and HPrEC cells. The effect of NaA-silane-PEG on number of genes involved in autoimmune and inflammatory responses (**A**) and in NF-κB signaling (**B**). The effect of NaA-silane-PEG-D2B on the number of genes involved in autoimmune and inflammatory responses (**C**) and in NF-κB signaling (**D**).

**Figure 5 ijms-22-05702-f005:**
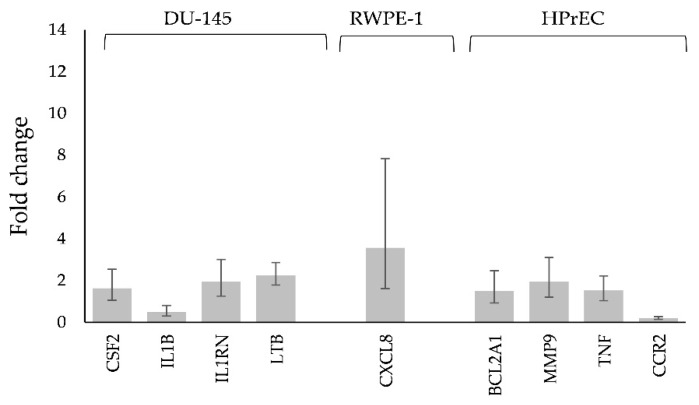
Statistically significant changes in NF-κB signaling and in the inflammatory response and autoimmunity gene expression with a cut-off value for gene expression fold changes >1.5 in DU-145 cells, RWPE-1 and HPrEC cells after treatment with 50 mcg mL^−1^ of NaA-silane-PEG for 96 h. Mean fold change values from three independent experiments are presented. Error bars represent a 95% confidence interval.

**Figure 6 ijms-22-05702-f006:**
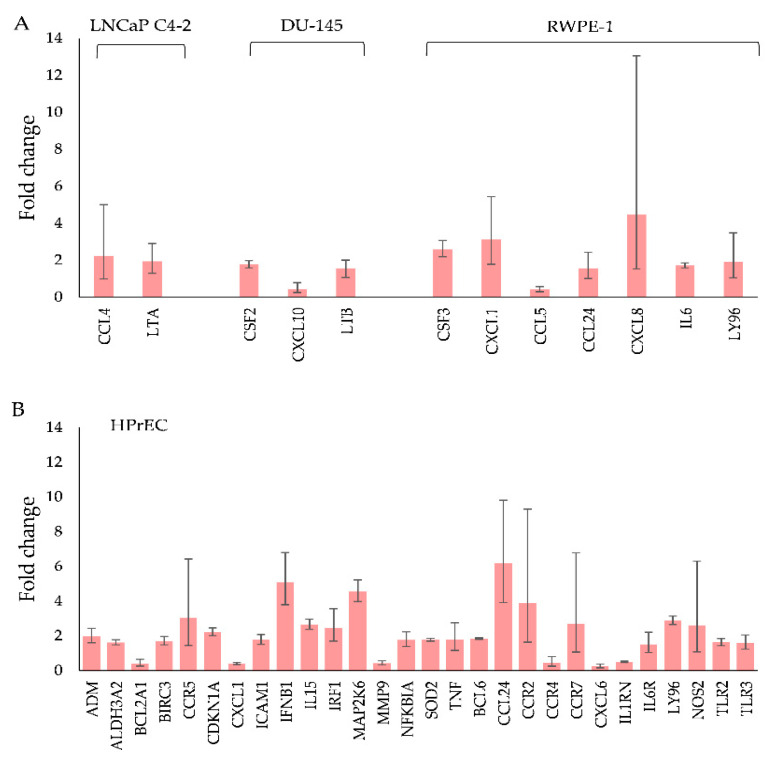
Statistically significant changes in NF-κB signaling and in the inflammatory response and autoimmunity gene expression with a cut-off value for gene expression fold changes >1.5 in (**A**) LNCaP C4-2, DU-145 cells, RWPE-1 and (**B**) HPrEC cells after treatment with 50 mcg mL^−1^ of NaA-silane-PEG-D2B for 96 h. Mean fold change values from three independent experiments are presented. Error bars represent a 95% confidence interval.

**Figure 7 ijms-22-05702-f007:**
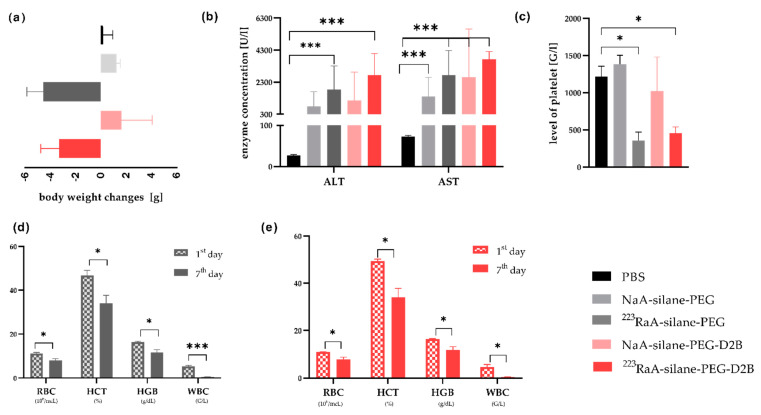
Influence of unlabeled and radiolabeled compounds on hematological and biochemical parameters of blood and body weight 7 days after a single injection of the tested compounds in BALB/c mice. Body weight (**a**), concentration of ALT and AST enzymes (**b**), concentration of platelets (**c**), hematological parameters in blood of mice exposed to NaA-silane-PEG (**d**) and NaA-silane-PEG-D2B (**e**). Data are expressed as the mean ± standard deviation (SD) from 10 mice/group. * *p* < 0.05, ** *p* < 0.01, *** *p* < 0.001 versus control group.

**Figure 8 ijms-22-05702-f008:**
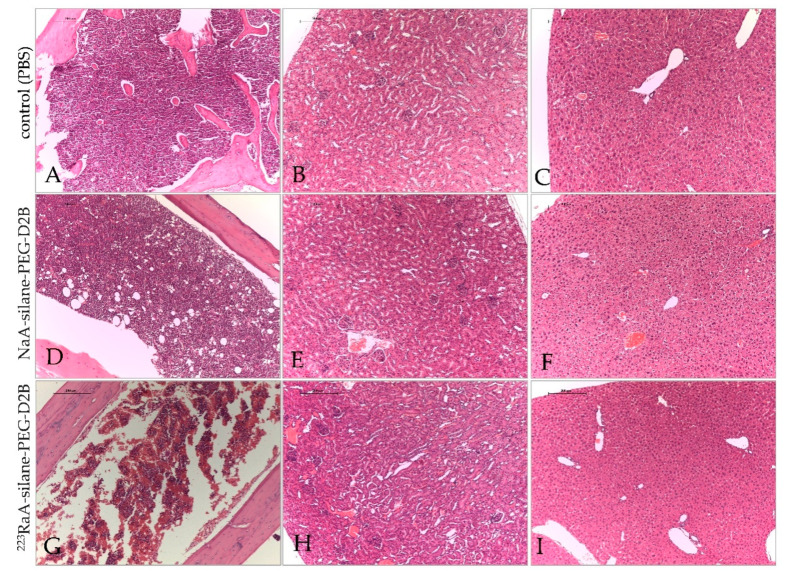
Representative images of bone marrow, kidneys and liver sections after exposure of BALB/c mice to NaA-silane-PEG-D2B and ^223^RaA-silane-PEG-D2B for 7 days. Bone marrow (**A**), kidneys (**B**) and liver (**C**) sections from control (PBS) mice. Bone marrow (**D**), kidneys (**E**) and liver (**F**) sections from mice after exposure to NaA-silane-PEG-D2B. Bone marrow (**G**), kidneys (**H**) and liver (**I**) sections from mice after exposure to ^223^RaA-silane-PEG-D2B. Scale bars, 200 mcm.

**Table 1 ijms-22-05702-t001:** Comparison of blood morphology data between first and seventh day after ^223^RaA-silane-PEG and ^223^RaA-silane-PEG-D2B injection into BALB/c mice. * statistically significant.

	^223^RaA-silane-PEG			^223^RaA-silane-PEG-D2B			Reference Values [25]
	1st Day	7th Day	*p*	1st Day	7th Day	*p*	
RBC (×10^3/^mcL)	11.1 ± 0.53	8.1 ± 0.71	0.030 *	11.0 ± 0.12	7.9 ± 0.86	0.030 *	6.93–12.24
HCT (%)	46.8 ± 2.26	34.0 ± 3.70	0.034 *	49.4 ± 0.86	34.1 ± 3.67	0.024 *	42.1–68.3
HGB (g/dL)	16.3 ± 0.40	11.7 ± 1.27	0.030 *	16.5 ± 0.15	11.9 ± 1.36	0.034 *	12.6–20.5
MCV (fL)	44.5 ± 0.55	42.9 ± 0.20	0.034 *	44.8 ± 0.23	43.1 ± 0.15	0.005 *	50.7–64.4
MCH (pg)	14.8 ± 0.21	14.7 ± 0.45	0.592	14.9 ± 0.12	14.9 ± 0.23	>0.999	13.2–17.6
MCHC (g/dL)	33.4 ± 0.35	34.7 ± 0.50	0.067	33.3 ± 0.38	34.7 ± 0.49	0.068	23.3–32.7
WBC (G/L)	5.4 ± 0.42	0.5 ± 0.06	0.000 *	4.6 ± 1.10	0.4 ± 0.10	0.028 *	3.48–14.03
NEUT (G/L)	1.1 ± 0.32	0.05 ± 0.03	0.030 *	1.1 ± 0.36	0.05 ± 0.04	0.039 *	-
LYMPH (G/L)	3.1 ± 0.67	0.3 ± 0.07	0.017 *	3.3 ± 0.92	0.3 ± 0.10	0.035 *	-
MONO (G/L)	0.2 ± 0.04	0.01 ± 0.00	0.011 *	0.1 ± 0.05	0.01 ± 0.01	0.078	-
BASO (G/L)	0.1 ± 0.01	0.02 ± 0.01	0.007 *	0.1 ± 0.02	0.01 ± 0.01	0.020 *	-
EOS (G/L)	0.01 ± 0.00	0.00 ± 0.00	0.067	0.01 ± 0.00	0.0 ± 0.00	0.088	-
PLATELET (G/L)	965.3 ± 104.00	355.3 ± 116.17	0.022 *	950.7 ± 183.07	457.7 ± 81.45	0.064	420–1698

**Table 2 ijms-22-05702-t002:** Biodistribution of ^223^RaA-silane-PEG and ^223^RaA-silane-PEG-D2B in BALB/c mice. Values are expressed as %ID g^−1^ and presented as mean ± standard deviation.

		**24 h**	**7 d**
**^223^** **RaA-silane-PEG**	Blood	2.68 ± 2.54	0.90 ± 0.44
	Lungs	18.13 ± 9.07	1 59 ± 1.48
	Liver	38.14 ± 2.22	20.06 ± 1.11
	Spleen	27.92 ± 4.77	40.71 ± 19.05
	Kidneys	2.46 ± 0.16	0.53 ± 0.09
	small intestine	0.25 ± 0.04	0.09 ± 0.01
	large intestine	1.13 ± 0.29	0.53 ± 0.16
	Stomach	0.59 ± 0.13	0.13 ± 0.08
	Bone	10.30 ± 1.13	19.52 ± 4.44
	Muscle	0.39 ± 0.14	0.16 ± 0.10
	urine [%ID]	14.75 ± 0.71	
	mass of spleen [mg]	70 ± 7	28 ± 10
		**24 h**	**7 d**
	Blood	5.90 ± 3.81	1.42 ± 0.58
**^223^** **RaA-silane-PEG-D2B**	Lungs	33.78 ± 8.10	2.28 ± 0.94
	Liver	57.34 ± 6.97	28.58 ± 0.91
	Spleen	69.34 ± 18.62	46.47 ± 18.16
	Kidneys	3.38 ± 0.02	0.80 ± 0.05
	small intestine	0.73 ± 0.85	0.08 ± 0.04
	large intestine	1.19 ± 0.50	0.17 ± 0.07
	Stomach	0.61 ± 0.08	0.10 ± 0.04
	Bone	17.74 ± 10.08	24.53 ± 3.87
	Muscle	0.33 ± 0.08	0.16 ± 0.07
	urine [%ID]	17.56 ± 1.71	
	mass of spleen [mg]	54 ± 15	35 ± 9

**Table 3 ijms-22-05702-t003:** Biodistribution of ^223^RaA-silane-PEG and ^223^RaA-silane-PEG-D2B in BALB/c Nude mice at 4, 24, 72 h and 7 days post injection. Values are expressed as %ID g^−1^ and presented as mean ± standard deviation.

**^223^** **RaA-silane-PEG**		**4 h**	**24 h**	**72 h**	**7 Days**
	Blood	3.13 ± 1.07	0.3 ± 0.19	2.62 ± 0.77	0.61 ± 0.30
	Lungs	61.50 ± 37.46	5.5 ± 5.27	14.75 ± 9.81	3.75 ± 2.11
	Liver	116.0 ± 28.45	28.0 ± 18.6	36.57 ± 11.48	31.46 ± 10.13
	Spleen	79.6 ± 22.26	15.2 ± 13.02	5.54 ± 4.28	49.69 ± 19.01
	Kidneys	5.8 ±2.16	1.3 ± 0.86	1.05 ± 0.41	0.69 ± 0.15
	small intestine	1.0 ± 0.20	0.2 ± 0.18	0.03 ± 0.01	0.10 ± 0.03
	large intestine	1.2 ± 0.26	0.7 ± 0.47	0.45 ± 0.12	0.13 ± 0.05
	Stomach	0.9 ± 0.45	0.2 ± 0.08	1.65 ± 1.97	0.10 ± 0.02
	Bone	4.2 ± 1.60	3.0 ± 1.87	10.62 ± 2.50	13.23 ± 4.85
	Muscle	0.5 ± 0.43	0.2 ± 0.11	0.12 ± 0.11	0.32 ± 0.12
	Tumor	0.71 ± 0.72	0.40 ± 0.18	0.05 ± 0.07	0.11 ± 0.06
		**4 h**	**24 h**	**72 h**	**7 Days**
**^223^** **RaA-silane-PEG-D2B**	Blood	1.0 ± 0.30	0.9 ± 0.41	4.11 ± 1.15	0.63 ± 0.39
	Lungs	20.2 ± 15.75	13.8 ± 13.51	33.80 ± 24.20	4.29 ± 1.58
	Liver	46.6 ± 7.11	62.5 ± 17.78	57.35 ± 8.68	43.55 ± 7.99
	Spleen	48.5 ± 5.24	36.6 ± 8.78	28.06 ± 24.74	128.57 ± 14.58
	Kidneys	2.3 ± 0.86	2.1 ± 0.37	1.81 ± 0.49	1.28 ± 0.29
	small intestine	0.6 ± 0.24	0.5 ± 0.09	0.09 ± 0.04	0.19 ± 0.10
	large intestine	0.7 ± 0.24	1.0 ± 0.66	0.61 ± 0.18	0.45 ± 0.22
	Stomach	0.3 ± 0.17	0.2 ± 0.10	1.29 ± 0.48	0.43 ± 0.35
	Bone	2.3 ± 1.42	5.7 ± 1.55	14.68 ± 6.40	15.02 ± 9.01
	Muscle	0.2 ± 0.07	0.3 ±0.19	0.27 ± 0.07	0.43 ± 0.16
	Tumor	0.23 ± 0.31	0.13 ± 0.19	0.24 ± 0.11	0.08 ± 0.03

## Data Availability

Not applicable.

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
