# Peer review of "Design and Evaluation of 223Ra-Labeled and Anti-PSMA Targeted NaA Nanozeolites for Prostate Cancer Therapy—Part II. Toxicity, Pharmacokinetics and Biodistribution"

_ijms, 2021, doi:10.3390/ijms22115702_

Round 1
Reviewer 1 Report
Interesting and extensive work.
However, must be more focused. Must be shortened by 50%. Remove in vitro data on gene expression effects in tumor cells.
Move all 133Ba data to supplement.
Focus on 223Ra biodistribution data in vivo and toxicity in mice. Should be compared with 223RaCl2 (free 223Ra) data. Use 223RaCl2 data from the litterature if needed.
Taken into account the physical half life of 11.4 days for 223Ra, why such a short follow up period in the toxicity study? Usually toxicity of radionuclides are seen after a few weeks to several months (e.g. check animal data for 223Ra (Xofigo)).
Check table 4. Spleen levels drops steadily until 72 h and then increases strongly at 7 days. Any explanation for this? Why was spleen levels much higher with 223Ra than with 133Ba? It could not all be accounted for by radiation effects to the spleen from 223Ra.
check spelling errors.
Reviewer 2 Report
The manuscript covers results related to in vitro cytotoxicity and in vivo toxicity and biodistribution evaluations of alfa-emitting Ra-labeled zeolite nanoparticles functionalized with PEG or PEG-targeting antibody. The manuscript is a follow up to a paper published in 2020 in the same journal where the synthesis and preliminary in vitro targetability results were presented. The study is carefully performed, and contains a lot of different type of data that suggest a relatively low, but cell-specific, in vitro toxicity of the particles. The in vivo data in terms of targetability and toxicity shows however that the particles are not suitable for in vivo applications, as the particles are taken up by the RES system and induces organ specific toxicity, and at the same time the particle uptake in the tumors remains very low, and is even lower for the antibody-tagged particles as compared to corresponding particles not carrying the targeting ligand. This slightly disappointing results is yet another demonstration for high in vitro promises of bionanosystems which cannot be reproduced in vivo because of short blood-half lives of the nanoparticles. It would have been advisable to study the uptake of the particles by macrophages in vitro before going through the lengthy in vivo characterizations, as it could simply be so that the PEGylation is simply not efficiently shielding the particles against opsonization. However, the study is honest and solid and should be accepted for publication after minor corrections: a) the amount of in vivo administered particles in g/kg should be given, not only the administered radioactivity. b) page 3, paragraph 2.1. The use of the word "markedly" to describe the small increase in apoptosis/necrosis of the antibody-tagged particles as compared to the PEGylated particles should be exchanged. If the numbers are 12.1% in one case and 13.5% in another, the difference is not markedly different, and it remains unclear if this difference is even statistically relevant. c) the limited stability of the conjugation of Ra and Ba to the zeolite has to be further highlighted in the discussion as a show stopper, as this will mean that the particles would not be clinically interesting even if the blood circulation time could be increased and the biodistribution enhanced.
Round 2
Reviewer 1 Report
Page 12-13, (lines 387-417). A brief discussion about the short follow-up of 7 days was due to animal welfare consideration and that the toxicity dosing was very high (~2 MBq/kg bw?) compared with clinical dosing of Xofigo (55 kBq/kg bw) should be included.
